# The Kinetics of Total Phenolic Content and Monomeric Flavan-3-ols during the Roasting Process of Criollo Cocoa

**DOI:** 10.3390/antiox9020146

**Published:** 2020-02-09

**Authors:** Editha Fernández-Romero, Segundo G. Chavez-Quintana, Raúl Siche, Efraín M. Castro-Alayo, Fiorella P. Cardenas-Toro

**Affiliations:** 1Programa Académico de Ingeniería Agroindustrial, Facultad de Ingeniería y Ciencias Agrarias, Universidad Nacional Toribio Rodríguez de Mendoza de Amazonas, Calle Higos Urco 342-350-356, Chachapoyas, Amazonas, Peru; fer.virgo59@gmail.com; 2Instituto de Investigación, Innovación y Desarrollo para el Sector Agrario y Agroindustrial de la Región Amazonas (IIDAA-Amazonas), Facultad de Ingeniería y Ciencias Agrarias, Universidad Nacional Toribio Rodríguez de Mendoza de Amazonas, Calle Higos Urco 342-350-356, Chachapoyas, Amazonas, Peru; segundo.quintana@untrm.edu.pe; 3Facultad de Ciencias Agropecuarias, Universidad Nacional de Trujillo, Av. Juan Pablo II s/n, Ciudad Universitaria, Trujillo 13001, Peru; rsiche@unitru.edu.pe; 4Sección de Ingeniería Industrial, Departamento de Ingeniería, Pontificia Universidad Católica del Perú, Av. Universitaria 1801, San Miguel 150136, Lima 32, Peru; fcardenas@pucp.pe

**Keywords:** roasting, catechin, epicatechin, total phenolic content, Criollo cocoa, kinetic

## Abstract

Cocoa beans are the main raw material for the manufacture of chocolate and are currently gaining great importance due to their antioxidant potential attributed to the total phenolic content (TPC) and the monomeric flavan-3-ols (epicatechin and catechin). The objective of this study was to determine the degradation kinetics parameters of TPC, epicatechin, and catechin during the roasting process of Criollo cocoa for 10, 20, 30, 40, and 50 min at 90, 110, 130, 150, 170, 190, and 200 °C. The results showed a lower degradation of TPC (10.98 ± 6.04%) and epicatechin (8.05 ± 3.01%) at 130 °C and 10 min of roasting, while a total degradation of epicatechin and a 92.29 ± 0.06% degradation of TPC was obtained at 200 °C and 50 min. Reaction rate constant (k) and activation energy ($E_{a}$) were 0.02–0.10 min^−1^ and 24.03 J/mol for TPC and 0.02–0.13 min^−1^ and 22.51 J/mol for epicatechin, respectively. Degradation kinetics of TPC and epicatechin showed first-order reactions, while the catechin showed patterns of formation and degradation.

## 1. Introduction

Cocoa beans, the seeds of the tree *Theobroma cacao* L., are the key raw material for chocolate production [1]. Criollo is the finest variety of cocoa [2]. Its seeds are aromatic, mild tasting, and with low bitterness: they represent the ideal raw material for a quality chocolate [1,3]. For obtaining chocolate from cocoa beans, one very important stage in the process is the roasting [4], which results in the production of desirable flavor and aroma compounds, as well as color changes [5]. The temperature and duration of roasting substantially affected the character in chemical and physical changes of cocoa beans [6].

Cocoa and chocolate have recently gained much attention due to their potential benefits in human health [1,7,8,9,10] and have recently become the target of increased scientific research due to their health promoting properties [11]. Cocoa, from a therapeutic viewpoint, is important due to the high concentrations of polyphenols as antioxidants [12]. These health effects have been assumed to be associated with the presence of polyphenols, among which are monomeric flavan-3-ols: epicatechin and catechin [5,13,14,15,16]. The beneficial effects of cocoa polyphenols to human health are, among others, the scavenge of free radicals and prevention of damage to DNA, the chelation of metals, vasoprotective effects, improvements in endothelial function, anti-inflammatory effects, the amelioration of insulin resistance, and anticarcinogenic effects, among others [13,17,18,19,20,21]. Cocoa is a rich source of polyphenolic compounds and may account for 12–18% of the dry mass of the beans [5]. Typically, polyphenols are sensitive to heat during the process, especially under a high temperature environment, i.e., roasting and drying [22,23]. Additionally, roasting influences the alteration of bioactive compounds [24]. In particular, the roasting process leads to the loss or modification of flavanols, which leads to a 14% loss of the total phenolic content (TPC), as well as the epimerization of epicatechin to catechin [25,26,27,28,29,30,31,32].

Kinetic modeling can provide a deeper understanding of the changes that occur during thermal processing controlling and food quality optimization [33,34]. Since these roasting techniques were introduced to the chocolate industry, the roasting time and temperature have been studied, applying designs or models that allow for a proper assessment of the process [35]. Various studies have been reported on the effect of roasting on cocoa nibs’ coloration, physical/chemical changes [4,5,8,16,23,25,36], flavor changes [37], and the kinetics of polyphenol degradation during drying [12,28,38]; however, studies on the kinetics of the polyphenol and monomeric flavan-3-ol degradation of Criollo cocoa during the roasting process are scarce, so the aim of the present study was to understand this degradation and determine the kinetic parameters.

## 2. Materials and Methods

### 2.1. Materials

Approximately 14 kg of dried fermented cocoa beans of Criollo variety were obtained directly from Multi-Service Cooperative APROCAM in the Bagua province, Amazon region, Perú.

### 2.2. Chemicals and Standards

Methanol HPLC grade (JT Baker, Deventer, The Netherlands), Folin–Ciocalteu’s phenol reagent, gallic acid, sodium carbonate, ≥98% (-)-epicatechin (HPLC) from green tea, ≥97% (-)-catechin (HPLC) from green tea, and ≥90% petroleum ether were purchased from Sigma Aldrich (Diessenhofen, Germany).

### 2.3. Roasting Process

Samples of cocoa bean were subjected to different treatments of time and temperature as conditions of the roasting process. Prior to the roast, the beans were selected according to their size, choosing beans of uniform size. Criollo cocoa samples (100 g) were roasted in a roaster (IMSA, ERTC-51, Lima, Perú) in the Agroindustry Plant Pilot at Universidad Nacional Toribio Rodríguez de Mendoza de Amazonas (UNTRM), Perú.

### 2.4. Chemical Analysis

#### 2.4.1. Methanolic Extraction of Phenolic Compounds and Monomeric Flavan-3-ols

According to Summa et al. [29] regarding some modifications, three cocoa beans were ground in a pestle from each sample. The powder was defatted by extraction with petroleum ether and centrifuged using a centrifuge (MPW Med Instruments, MPW-51, Warszawa, Poland) at 3000 rpm for 15 min at room temperature. The supernatant was then discarded. Fresh petroleum ether was added and then centrifuged (four times). The resulting defatted material was air dried at room temperature. Methanol extraction based on the methodology used by Jonfia-Essien et al. [30] with some modifications was performed, and 0.5 g of defatted cocoa powder was homogenized in 25 mL of 80% methanol for 30 min in a magnetic stirrer and then filtered in a vacuum filter. This methanolic extract was used for the determination of both TPC and monomeric flavan-3-ols.

#### 2.4.2. Total Phenolic Content

The TPC of the Criollo cocoa beans was determined following the procedures of Singleton et al. [31] and Hu et al. [7]. Diluted extract (0.1 mL) or blank (0.1 mL deionized water) was mixed with 7.9 mL water and 0.5 mL of Folin-Ciocalteu reagent for 5 min at 22 °C. Next, 1.5 mL of saturated sodium carbonate solution was added. Reagents were mixed thoroughly by vigorous shaking for 10 s by hand. The mixture was incubated in the dark at 22 °C for 2 h before determination of the absorbance at 765 nm using an UV/Visible spectrophotometer (Unico, S2100, Dayton, NJ, USA). Gallic acid in 70% methanol was diluted (2–16 mg/L) to create a calibration curve. TPC is expressed as mg of gallic acid equivalents/g defatted cocoa bean (mg GAE/gdf).

#### 2.4.3. Quantification of Epicatechin and Catechin of the Methanolic Extract

Quantification of epicatechin and catechin followed the procedures of Wang et al. [39] with some modifications in a high performance liquid chromatography system (HPLC) (Lachrom Elite, Hitachi, Japan) equipped with a UV-Vis detector (Hitachi, Lachrom Elite L–2420, Wako, Japan) and isocratic pump (Hitachi, Lachrom Elite L–2130, Japan). The column used was a C18 150 × 4.6 mm, 5 µm (Merck, Purospher RP–18 endcapped, Darmstadt, Germany). The mobile phase was methanol/water/orthophosphoric acid (20/79.9/0.1), and the flow rate was 1 mL/min. Absorption wavelength was selected at 210 nm. The sample injection volume was 20 µL. Chromatographic peaks in the samples were identified by comparing their retention time and UV spectrum with those of the reference standards. A standard graph for each component was prepared by plotting concentration versus area. Quantification was carried out from integrated peak areas of the sample and corresponding standard graphs, and the epic/cat ratios of epicatechin and catechin concentrations were obtained.

#### 2.4.4. TPC and Monomeric Flavan-3-ol Degradation

According to Martins et al. [40], the degradation was then calculated according to the following formula:(1)$$\%\;degradation=\left(\frac{X_{control}-X_{t}}{X_{control}}\right)*100$$
where $X_{control}$ is the TPC, epicatechin, or catechin concentration in the control sample (unroasted cocoa bean), and $X_{t}$ is the TPC, epicatechin, or catechin concentration at time *t*.

### 2.5. Experimental Desing for Kinetics of TPC and Monomeric Flavan-3-ols

Samples (100 g) in triplicate of Criollo cocoa beans were roasted for each combination of time and temperature (10, 20, 30, 40, and 50 min at 90, 110, 130, 150, 170, 190, and 200 °C), 105 samples were obtained. Means and standard deviations were calculated. The temperature and time values used were defined according to the published literature. Fitting procedures (Matlab 2014) were used to determine the reaction rate constants for TPC, epicatechin, and catechin. The general kinetic models of zero-, first-, and second-order reactions were used, presented in Equations (2)–(4), respectively [41].
(2)$${\left[A\right]}_{0}-\left[A\right]=kt$$
(3)$$\left[A\right]={\left[A\right]}_{0}\;exp\left(-kt\right)$$
(4)$$\frac{1}{\left[A\right]}-\frac{1}{{\left[A\right]}_{0}}=\;kt$$
k (min^−1^) is the reaction rate constant at temperature T; t is the reaction time (min); ${\left[A\right]}_{0}$ and [A] are the initial (control sample) and final amounts of TPC, epicatechin or catechin, respectively, at different times t and temperatures.

The final fitting models were obtained by matching and analyzing the initial models obtained by Matlab software with general equations of zero-, first-, and second-order reactions presented in Equations (2)–(4), respectively. The effect of temperature was evaluated by means of the Arrhenius equation.
(5)$$k=k_{0}*e^{\left(-E_{a}/RT\right)}.$$

The activation energy $E_{a}$ for the formation of each parameter was determined by linear regression of Ln k  curve versus 1/T with Equation (6).
(6)$$Lnk=Lnk_{0}-\;E_{a}/RT$$
$E_{a}$ (J/mol) is the apparent activation energy, R (8.3145 J/mol.K) is the universal gas constant, k is the reaction rate constant, and $k_{0}$ is the pre-exponential factor.

### 2.6. Statistical Analysis

The results were compared using one-factor analysis of variance (ANOVA) followed by the Tukey test. Previously, Dixon’s Q test, for the identification and rejection of outliers, was used. Statistical analyses were carried out in Minitab 17 software.

## 3. Results

### 3.1. Effect of Roasting on Monomeric Flavan-3-ols and TPC

The TPC, epicatechin, and catechin concentration and the epi/cat ratio of the unroasted (control sample) Criollo cocoa beans were 110.98 ± 1.43 mg GAE/gdf, 30.29 ± 1.0 mg/gdf, 2.71 ± 0.13 mg/gdf, and 11.20 ± 0.40, respectively (Table 1). These values served as a starting point for the study of degradation. The TPC and epicatechin values and the epi/cat ratios of the treatments were lower than the control sample. Some results show that the concentration of epicatechin in some cases increased. During the roasting process, the epi/cat ratio was reduced as the process temperature increased. Considering a time of 10 min, the epi/cat ratio was reduced from 6.08 ± 0.96 when heated at 90 °C to 3.20 ± 0.68 at 200 °C. Table 1 shows that epicatechin degradation reached 100% when Criollo cocoa beans were roasted for 50 min at 190 or 200 °C, while TPC did not reach full degradation at any temperature or time. The opposite happened with catechin, which showed patterns of formation. The values in bold indicate that there was formation instead of degradation.

### 3.2. Roasting Kinetics of Monomeric Flavan-3-ols and TPC

The values obtained from the fitted parameters are given in Table 2. In the case of TPC and epicatechin, the k value increases from 0.02 ± 0.01 min^−1^ at 90 °C to 0.10 ± 0.05 min^−1^ at 200 °C, and from 0.02 ± 0.01 min^−1^ at 90 °C to 0.13 ± 0.04 min^−1^ at 200 °C, respectively. The kinetic parameters of the catechin are not shown because it corresponds to a combined production and degradation model, which does not correspond to Equations (2), (3), or (4).

Figure 1 shows that the catechin reaction kinetics could be divided into two order models: one for formation and the other for degradation, where the highest catechin formation (**84.90 ± 37.90%**) was at 170 °C and 30 min of roasting.

The temperature dependence of the TPC and epicatechin degradation was estimated using the Arrhenius equation expressed in Equation (5). The linear behavior of *Ln*
k versus 1/T allowed us to determine $E_{a}$ for TPC and epicatechin of 24.03 J/mol (R^2^ = 0.94) and 22.51 J/mol (R^2^ = 0.97), respectively (Figure 2). The parameters for catechin kinetics were not calculated because it presented a degradation and production pattern, as mentioned above.

## 4. Discussion

The time and temperature of the roasting process depend on several factors, such as the type of cocoa (Criollo or Forastero) and others [42]. The main type of polyphenols (known for their demonstrated antioxidant and anti-inflammatory properties) in cocoa is flavanols. This family of compounds includes catechin and epicatechin (monomeric species). Epicatechin is the most abundant flavanol in cocoa and accounts for 35% of the total polyphenolic fraction [43,44] (TPC). In one study made by Kim and Keeney [45], the epicatechin concentrations ranged from 2.66 (Jamaica) to 16.52 mg/g (Costa Rica) of the defatted sample in cocoa beans of different varieties. The epicatechin and catechin concentrations of the control sample were higher than those found by Kim and Keeney [45] and Payne et al. [32] in fermented cocoa beans (the Forastero variety) from Ivory Coast (epicatechin: 1.69 ± 0.10 mg/g; catechin: 0.08 ± 0.00 mg/g) and Papua New Guinea (epicatechin: 0.78 ± 0.04 mg/g; catechin: 0.05 ± 0.00 mg/g). These values were higher than those found by Mazor Jolić et al. [46] in Forastero cocoa beans from Ghana (epicatechin: 2.23 ± 0.6 mg/g; catechin: 0.28 ± 0.04 mg/g). Other studies conducted in Perú on fermented cocoa beans from Tingo María, San Alejandro, and Curimaná presented epicatechin and catechin concentrations of 0.33–5.04 mg/g and 0.02–0.14 mg/g, respectively [47]. It is known that epicatechin has diverse biological properties (antioxidant, antimicrobial, anti-inflammatory, antitumor, and cardio-protective activity) [48]; thus, the Criollo cocoa beans used in the present study show potential for use in the elaboration of functional foods. It has been postulated that the ratio of epicatechin to catechin (epi/cat) possibly could be associated with the degree of cocoa processing [32,49]. Payne et al. [32] obtained epi/cat ratio values of 20.1 ± 0.63 for dry fermented cocoa beans and 3.35 ± 0.20 at 90 °C to 0.96 ± 0.02 at 120 °C for roasted beans. Table 1 shows control sample epi/cat ratio values of 11.20 ± 0.40; considering 50 min of roasting, values of 5.55 ± 1.24 at 90 °C to 2.36 ± 0.20 at 130 °C and 0.00 ± 0.00 at 200 °C were obtained. These results demonstrate that roasting at 200 °C at any time (10 or 50 min) is aggressive for epicatechin, so if these parameters are used, the functional properties of chocolate will be lost.

In Table 1, a roasting time of 50 min at 90, 110, 130, 150, 170, 190, and 200 °C degraded the TPC to 53.46 ± 7.29, 46.88 ± 6.30, 65.40 ± 3.16, 74.57 ± 1.37, 82.40 ± 0.89, 88.91 ± 1.27, and 92.29 ± 0.06, respectively; these results are consistent with those obtained by Mazor Jolić et al. [46]. The decrease of TPC is associated with the thermal and oxidative degradation of these compounds [35]. It was proven that, at high temperature, low molecular weight phenolic compounds easily volatilize [50]. Djikeng et al. [50] also observed that the degradation percentage of the TPC also increased with roasting time. In general, more intense roasting conditions result in a greater loss of TPC due to the high redox activity of polyphenols at those conditions [50,51]. Epicatechin degraded more than TPC at a higher temperature, obtaining a complete degradation at 200 °C and 50 min. This is concordant with Stanley et al. [5], who observed the significant effects of roasting time within temperatures up to 190 °C; the levels of epicatechin in Trinitario cocoa (a type of fine cocoa) decreased in a time- and temperature-dependent manner. Significant decreases in Criollo cocoa were observed. Among the monomeric flavan-3-ols, epicatechin was identified as the more active compound responsible for the vascular health benefits associated with cocoa and chocolate [52]. Applying treatments of 190–200 °C results in a total loss of epicatechin, and the healthy properties of the Criollo cacao and its aromatic properties may also be reduced; for this reason, Żyżelewicz et al. [1] and Hurst et al. [36] state that Criollo beans require milder roasting conditions. The roasting temperature is generally in the range from 110 to 120 °C.

The most important reactions occurring with catechins under thermal processing are epimerization, hydrolysis, oxidation, and polymerization; catechin epimerization takes place on two asymmetric carbon atoms in the C ring [53]. In Table 1, catechin shows a particular behavior due to the increase in its concentration (formation) at the beginning of the roasting. The catechin values in bold font shows an increase in its concentration with respect to the control sample (unroasted beans) during the first 10 min at 130 °C (**11.46 ± 4.71%**), and its formation continues until the first 10 min at 200 °C and subsequently degrades to 21.37 ± 5.74%. The trend of these results is consistent with those obtained by Stanley et al. [5], except that the catechin formation was much higher, showing an increase in catechin concentration 675% higher than the control (unroasted beans) at 10 min. These results were reported for Trinitario cocoa beans. We can affirm that, in Trinitario cacao, the formation of catechin is much greater than in Criollo cacao, and this applies to the epimerization of epicatechin to catechin [5,14,25,32,54]; however, as the treatment continued, its concentration decreased. These results were also found by Żyżelewicz et al. [4] in Forastero cocoa, since after 15 min of roasting at 135 °C, the content of catechin increased significantly, initiating its degradation. When cocoa beans are progressively roasted at conditions described as low, medium, and high roast conditions (160 °C at 13, 20, and 25 min), there is a progressive loss of epicatechin and an increase in catechin with higher roast levels [36,55]. The high temperature–short time (HTST) process induces higher epicatechin epimerization than does the low temperature–long-time (LTLT) process, generating greater amounts of catechin [16,53].

Cocoa consumption is suggested to promote health benefits. The amounts and profiles of monomeric flavanols depend strongly on the bean type, origin, and manufacturing process. Roasting is known as a crucial step in the technical treatment of cocoa, which leads to flavanol losses and modifications, especially the epimerization of epicatechin to catechin [42]. These modifications were fully produced (epimerization) at temperatures of 150, 170, and 190 °C (Table 1). In the manufacture of chocolate, it is necessary to achieve a balance that maintains healthy properties and the development of the aroma that characterizes fine, Criollo cocoa chocolates.

In order to explain the phenomena of TPC and epicatechin degradation, the data were fitted using kinetic models, which are pseudo first-order kinetic reactions coinciding with the majority of food reactions [56]. The term “pseudo” is added with respect to reactions in biological materials, such as food, because the actual reaction mechanism and its kinetics are far more complex [28]. We used seven temperatures in our experiment, allowing us to notice the behavior of the roasting process; when Taoukis et al. [56] states that five or six experimental temperatures are the practical optimum to obtain meaningfully narrow confidence limits in kinetic parameters ($E_{a}$ and k). The values obtained from the fitted parameters in Matlab are given in Table 2 and consistent with previous studies [26,28,41,57], in which the first-order kinetic model explains both TPC and epicatechin degradation at all temperatures; the k value of epicatechin is higher than the k value of TPC. These results suggest that the degradation of epicatechin is faster than the degradation of TPC. Roasting conditions of 130 °C and 10 min produced a lower percentage of epicatechin (8.05 ± 3.01%) and TPC (10.98 ± 6.04%) degradation. Many authors claim that these compounds are degraded by heat [4,22,23,50]. The most intense temperatures have caused significant degradation of the components; however, it was found that the epicatechin is more thermosensitive than TPC. This is explained by the greater k value that epicatechin has compared to TPC; however, the chemical basis of this phenomenon must be studied.

Kinetic modeling may also use the influence of processing on critical quality parameters. Knowledge of degradation kinetics, including the reaction order, $E_{a}$ and k, is of great importance to predict food quality loss during thermal process treatments [58]. Food quality loss reactions described by the kinetic models were shown to follow Arrhenius (Equation (5)) behavior with temperature changes [56]. The $E_{a}$ for TPC was greater than the $E_{a}$ for epicatechin, this being the minimal value of energy that a specific collision between reagent molecules must achieve in order for a reaction to take place [59]. The epicatechin thus needs less energy than TPC to degrade it. The relative differences in $E_{a}$ values could be due to the different composition of the sample studied or to changes occurring in the samples during heating [58]. Olivares-Tenorio et al. [60] observed that catechin followed patterns of formation by epimerization and degradation, and experiments need to be devised to unravel these various reactions. Figure 1 shows that catechin reaction kinetics could be divided into a two order models: one for formation and the other for degradation, where the highest catechin formation was at 170 °C and 30 min (**84.90 ± 37.90%**), followed by treatments at 190 °C and 150 °C. The largest catechin formation occurs in the first 30 min of roasting at any of the aforementioned temperatures. The epimerization from epicatechin to catechin due to technological treatment (heat) has often been postulated. Only a few publications have confirmed this reaction by enantioseparation. The reaction mechanism is not fully clarified, but it is assumed that ring opening occurs at position C-2 of the oxygenated ring, and reclosing leads to the atypical enantiomers. High temperatures, particularly when combined with alkaline conditions, accelerate the epimerization reaction [42]. Figure 1 shows that at 90 and 110 °C there is only catechin degradation; subsequently, the formation of catechin begins slowly from 130 °C, reaching its highest point at 190 °C. The treatment at 200 °C only shows a degradation pattern. It is assumed that the k for catechin formation kinetics will be greater at 190 °C for 30 min, after which the degradation kinetics will take place at a high k. An enantoseparation reaction will occur to a greater extent for 30 min when the Criollo cocoa beans are roasted at 150, 170, or 190 °C.

## 5. Conclusions

The roasting process of Criollo cocoa allowed for a higher formation of catechin in the first 30 min of the process at 170 °C, followed by degradation to minimum levels at 200 °C, while the epicatechin showed greater susceptibility to treatment than the TPC and the catechin. Likewise, the degradation data of TPC and epicatechin are better suited to a first-order kinetic reaction as the temperature increases. Although it is true that the roasting stage is essential for the development of the aromas of chocolate, this also affects the polyphenolic content and monomeric flavan-3-ol of cocoa beans and thus the final chocolate product; therefore, roasting at moderate temperatures is necessary to obtain minimal degradation of cocoa phenolic compounds and consequently antioxidant properties. Likewise, with epicatechin being part of the total phenolic compounds, its degradation percentage is higher. This is explained by its kinetics; however, its chemical explanation for future work is pending.

## Figures and Tables

**Figure 1 antioxidants-09-00146-f001:**
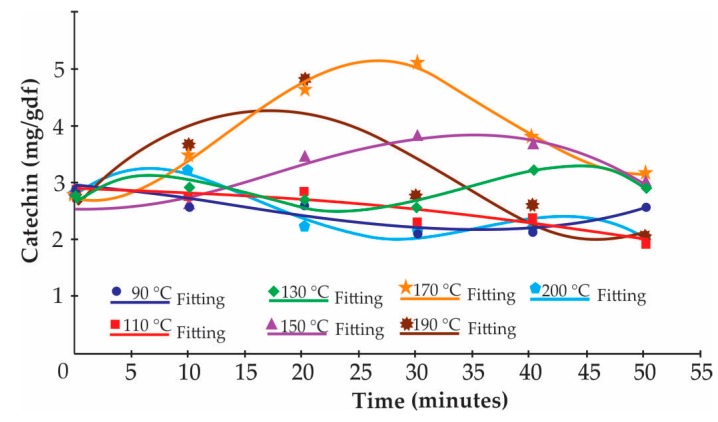
Kinetic of formation and degradation of catechin.

**Figure 2 antioxidants-09-00146-f002:**
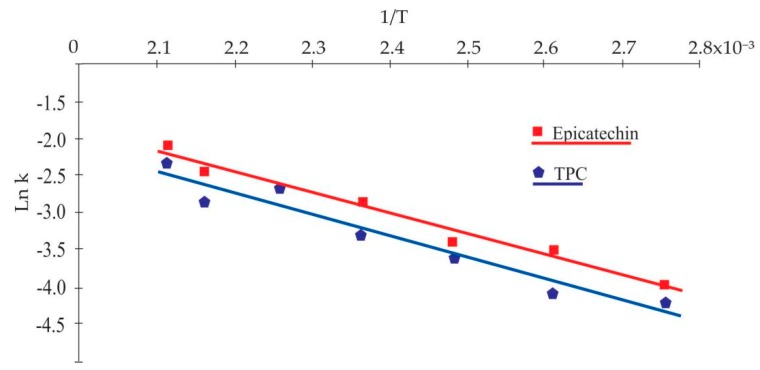
Correlation of *Ln*
k with 1/T to obtain the TPC and epicatechin kinetic parameters.

**Table 1 antioxidants-09-00146-t001:** Concentration and degradation of TPC and monomeric flavan-3-ols, epi/cat ratios, and the formation of catechin.

T (°C)	Time (min)	Concentration ^1^			Epi/Cat Ratio ^1^	Degradation ^1^ (%)		
		TPC (mg GAE/gdf)	Epicatechin (mg/gdf)	Catechin (mg/gdf)		TPC	Epicatechin	Catechin ^2^
Control	0	110.98 ± 1.43^a^	30.29 ± 1.09^a^	2.71 ± 0.13^a^	11.20 ± 0.40^a^			
90	10	54.60 ± 10.86^b^	14.97 ± 1.45^b^	2.48 ± 0.14^ab^	6.08 ± 0.96^b^	50.84 ± 9.47^a^	50.41 ± 6.61^a^	8.31 ± 5.43^a^
	20	72.70 ± 7.44^b^	17.78 ± 3.13^b^	2.48 ± 0.05^ab^	7.20 ± 1.37^b^	34.44 ± 7.46^a^	41.25 ± 10.23^a^	8.31 ± 6.24^a^
	30	45.10 ± 7.61^b^	11.78 ± 1.22^b^	2.20 ± 0.11^b^	5.35 ± 0.32^b^	59.41 ± 6.47^a^	61.07 ± 4.13^a^	18.69 ± 3.99^a^
	40	62.10 ± 19.80^b^	12.92 ± 4.89^b^	2.22 ± 0.28^b^	5.93 ± 2.62^b^	43.90 ± 18.7^a^	57.0 ± 17.7^a^	18.25 ± 6.22^a^
	50	51.58 ± 7.44^b^	13.75 ± 2.92^b^	2.48 ± 0.04^ab^	5.55 ± 1.24^b^	53.46 ± 7.29^a^	54.78 ± 8.05^a^	8.08 ± 5.82^a^
110	10	68.54 ± 4.05^c^	27.24 ± 0.53^c^	2.60 ± 0.07^cd^	10.50 ± 0.36^c^	38.21 ± 4.44^c^	10.02 ± 2.08^d^	2.22 ± 1.45^d^
	20	60.84 ± 5.59^cd^	15.36 ± 0.47^d^	2.67 ± 0.21^c^	5.78 ± 0.50^d^	45.14 ± 5.72^bc^	49.20 ± 3.07^c^	7.78 ± 2.00^cd^
	30	52.25 ± 3.86^d^	11.47 ± 0.46^e^	2.28 ± 0.08^de^	5.05 ± 0.28^de^	52.91 ± 3.59^b^	62.10 ± 1.50^b^	15.79 ± 4.65^bc^
	40	48.14 ± 6.12^d^	8.28 ± 1.36^e^	2.33 ± 0.18^cde^	3.54 ± 0.31^f^	56.58 ± 5.95^b^	72.53 ± 5.49^b^	13.59 ± 8.53^bcd^
	50	58.92 ± 6.57^cd^	8.22 ± 2.37^e^	2.06 ± 0.12^e^	3.96 ± 0.97^ef^	46.88 ± 6.30^bc^	72.77 ± 8.31^b^	24.00 ± 3.58^b^
130	10	98.85 ± 7.79^e^	28.74 ± 0.94^f^	3.01 ± 0.03^f^	9.55 ± 0.39^g^	10.98 ± 6.04^f^	8.05 ± 3.01^e^	**11.46 ± 4.71^e^**
	20	52.24 ± 3.68^f^	11.21 ± 1.24g	2.69 ± 0.47^f^	4.20 ± 0.26^h^	52.93 ± 3.24^e^	62.87 ± 5.25^d^	14.22 ± 6.08^e^
	30	35.34 ± 2.19^g^	9.47 ± 1.01^gh^	2.62 ± 0.23^f^	3.66 ± 0.67^hi^	68.14 ± 2.34^d^	68.77 ± 2.57^cd^	5.30 ± 4.23^e^
	40	44.35 ± 7.73^fg^	10.46 ± 1.46^g^	3.30 ± 0.36^f^	3.23 ± 0.81^hi^	60.01 ± 7.15^de^	65.40 ± 5.08^d^	**18.62 ± 7.50^e^**
	50	38.43 ± 3.98^fg^	6.98 ± 1.01^h^	2.95 ± 0.22^f^	2.36 ± 0.20^i^	65.40 ± 3.16^de^	76.99 ± 2.78^c^	**9.08 ± 7.50^e^**
150	10	46.71 ± 7.84^h^	11.25 ± 2.11^i^	2.62 ± 0.24^g^	4.26 ± 0.41^j^	57.85 ± 7.61^h^	62.70 ± 8.19^h^	8.66 ± 0.15^f^
	20	48.78 ± 8.12^h^	8.70 ± 0.70 ^i j^	3.49 ± 0.68^g^	2.59 ± 0.73^k^	56.07 ± 7.02^h^	71.29 ± 1.29^gh^	**32.90 ± 25.60^f^**
	30	31.32 ± 2.25^i^	6.32 ± 0.53^jk^	3.79 ± 0.75^g^	1.70 ± 0.25^kl^	71.80 ± 1.72^g^	79.14 ± 1.51^fg^	**39.60 ± 21.10^f^**
	40	31.27 ± 2.17^i^	5.72 ± 0.12^k^	3.64 ± 0.10^g^	1.57 ± 0.07^kl^	71.81 ± 2.18^g^	81.11 ± 0.84^fg^	**34.61 ± 8.41^f^**
	50	28.22 ± 1.32^i^	4.27 ± 0.68^k^	3.01 ± 0.35^g^	1.41 ± 0.08^l^	74.57 ± 1.37^g^	85.95 ± 1.81^f^	**11.05 ± 8.82^f^**
170	10	42.10 ± 7.56^j^	12.33 ± 2.33^l^	3.48 ± 0.98^hij^	0.08 ± 0.02^no^	62.12 ± 6.39^j^	59.44 ± 6.36^k^	**11.58 ± 0.01^h^**
	20	33.43 ± 7.28^jk^	6.75 ± 0.77^m^	4.66 ± 0.31^hi^	0.10 ± 0.02^n^	69.84 ± 6.81^ij^	77.75 ± 1.80^j^	**72.34 ± 12.07^gh^**
	30	31.50 ± 6.43^jk^	5.41 ± 0.69^mn^	5.15 ± 1.11^h^	0.17 ± 0.05^m^	71.66 ± 5.50^ij^	82.11 ± 2.36^ij^	**84.90 ± 37.90^g^**
	40	31.33 ± 7.40^jk^	4.16 ± 0.59^mn^	3.85 ± 0.66^hij^	0.12 ± 0.02^mn^	71.82 ± 6.36^ij^	86.26 ± 1.77^ij^	**43.00 ± 30.30^gh^**
	50	19.54 ± 1.05^k^	3.12 ± 0.18^n^	3.19 ± 0.42^ij^	0.11 ± 0.01^mn^	82.40 ± 0.89^i^	89.70 ± 0.77^i^	**22.19 ± 14.41^h^**
190	10	41.13 ± 8.69^l^	11.61 ± 4.88^o^	3.67 ± 0.91^l^	3.35 ± 1.62^p^	62.95 ± 7.70^l^	61.93 ± 15.42^m^	**42.3 ± 29.40^ij^**
	20	38.17 ± 6.69^l^	5.70 ± 0.50^p^	4.87 ± 0.28 ^k^	1.18 ± 0.17^q^	65.63 ± 5.75^l^	81.20 ± 0.99^l^	**80.50 ± 17.40^i^**
	30	21.35 ± 4.45^m^	3.11 ± 0.60^pq^	2.77 ± 0.25^lm^	1.11 ± 0.12^q^	80.79 ± 3.80^k^	89.70 ± 2.32^kl^	**7.32 ± 6.32^j^**
	40	19.81 ± 2.11^m^	2.43 ± 0.13^pq^	2.67 ± 0.27^lm^	0.92 ± 0.10^q^	82.17 ± 1.72^k^	91.97 ± 0.23^kl^	**9.97 ± 3.21^j^**
	50	12.31 ± 1.36^m^	0.00 ± 0.00^q^	2.05 ± 0.07^m^	0.00 ± 0.00^q^	88.91 ± 1.27^k^	100.00 ± 0.00^k^	23.96 ± 6.02^j^
200	10	33.23 ± 2.44^n^	8.10 ± 0.88^r^	3.20 ± 0.68^n^	2.64 ± 0.83^r^	70.04 ± 2.60^o^	73.21 ± 3.58^p^	**21.89 ± 14.12^k^**
	20	17.33 ± 3.26^o^	2.42 ± 0.21^s^	2.30 ± 0.30^n^	1.06 ± 0.05^s^	84.38 ± 2.98^n^	92.01 ± 0.39^o^	15.17 ± 7.86^k^
	30	13.76 ± 0.53^op^	2.32 ± 0.06^s^	2.26 ± 0.07^n^	1.03 ± 0.03^s^	87.60 ± 0.64^mn^	92.34 ± 0.10^o^	16.56 ± 1.54^k^
	40	14.41 ± 1.86^o^	2.53 ± 0.45^s^	2.35 ± 0.60^n^	1.09 ± 0.09^st^	87.00 ± 1.85^mn^	91.61 ± 1.81^o^	22.29 ± 10.34^k^
	50	8.56 ± 0.10^p^	0.00 ± 0.00^t^	2.13 ± 0.21^n^	0.00 ± 0.00^t^	92.29 ± 0.06^m^	100.00 ± 0.00^n^	21.37 ± 5.74^k^

^1^ Different letters in the same column represent statistically significant differences (*p* ≤ 0.05). At least three replicate samples were analyzed. ^2^ Bold values indicate an increase in catechin concentration.

**Table 2 antioxidants-09-00146-t002:** First-order kinetic parameters fitted for TPC and epicatechin degradation.

Roasting Temperature (°C)	TPC			Epicatechin		
	k **(min^−1^)**	$R^{2}$	RMSE	k **(min^−1^)**	$R^{2}$	RMSE
90	0.02 ± 0.01	0.52	18.50	0.02 ± 0.01	0.65	4.57
110	0.02 ± 0.01	0.70	13.97	0.03 ± 0.01	0.95	2.49
130	0.03 ± 0.02	0.87	13.08	0.03 ± 0.02	0.86	4.31
150	0.04 ± 0.03	0.83	14.59	0.06 ± 0.03	0.91	3.27
170	0.07 ± 0.03	0.96	2.24	0.07 ± 0.03	0.96	2.24
190	0.06 ± 0.03	0.92	11.35	0.09 ± 0.02	0.99	1.05
200	0.10 ± 0.05	0.96	9.23	0.13 ± 0.04	0.99	1.44

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
