# Peer review of "The Kinetics of Total Phenolic Content and Monomeric Flavan-3-ols during the Roasting Process of Criollo Cocoa"

_antioxidants, 2020, doi:10.3390/antiox9020146_

Round 1

Reviewer 1 Report

The research paper of Fernández-Romero et al. reports a study concerning the effect of temperature and time on the degradation pattern of phenolic compound contained in cocoa (Kinetics of total phenolic content and monomeric 2 flavan-3-ols during roasting process of Criollo cocoa).

The research is not particularly new, but contains supplementary information on a specific variety of cocoa. The experimental design is appropriated and the data are presented and discussed with clarity, although presentation in the Table and Figures are sometimes redundant.

Comments

Line 23: Is Matlab relevant for the abstract? Instead, a short description of the equation parameters can help the readers.

Line 41: not the chocolate, really. The literature cited does not support the effect of cocoa on human health; find more appropriated publications.

Line 49: the literature cited does not support the effect of cocoa on human health; find more pertaining publications.

Line 77: this is the mean value or all the beans had exactly the same size? Please clarify and if this is the mean value report also the s.d.

Line 79: Country?

Line 91: “Total phenolic content of methanol extract”

Line 101: “Quantification of epicatechin and catechin of the methanol extract”

Line 119: It is not clear if the same 100 g were subjected at the different times of roasting (sampling at each time an aliquot) or if for every time 100 g of cocoa were roasted.

Table 2: The kinetic of catechin is not reported in the table; add in the text why.

Figure 1A and 1B are redundant with the data presented in Table 1 and, in some way, with kinetic parameters of Table 2 and can be deleted. Figure 1C reports only the fitting of 110°C, 190°C and 200°C: which was the reason of this choice?

Figure 2: if R2 of the 90°C is excluded from the fitting, epicatechin is parallel to x axis and probably also TPC is not significant. The different pattern of 90°C is also evident in Table 1 and the Figure 2 can be deleted and text in the discussion reworded accordingly (Lines 242-244).

Conclusion: the authors did not measure the antioxidant activity and the sentence must be reworded.

Author Response

Response to Reviewer 1 Comments

Point 1. Line 23: Is Matlab relevant for the abstract? Instead, a short description of the equation parameters can help the readers.

Response 2: Matlab was deleted

Point 2. Line 41: not the chocolate, really. The literature cited does not support the effect of cocoa on human health; find more appropriated publications.

Response 2: More references were added about the potential benefits of chocolate in human health

Point 3. Line 49: the literature cited does not support the effect of cocoa on human health; find more pertaining publications.

Response 3: More references were added about the potential benefits of cocoa polyphenols in human health

Point 4. Line 77: this is the mean value or all the beans had exactly the same size? Please clarify and if this is the mean value report also the s.d.

Response 4. This is not a mean value

Point 5. Line 79: Country?

Response 5. UNTRM is in Perú

Point 6. Line 91: “Total phenolic content of methanol extract”

Response 6. The text changed to: “Methanolic extraction of phenolic compounds and monomeric flavan-3-ols” and “This methanolic extract was used for the determination of both TPC and monomeric flavan-3-ols”.

Point 7. Line 101: “Quantification of epicatechin and catechin of the methanol extract”

Response 7. The text changed to: “Quantification of epicatechin and catechin of the methanolic extract”

Point 8. Line 119: It is not clear if the same 100 g were subjected at the different times of roasting (sampling at each time an aliquot) or if for every time 100 g of cocoa were roasted.

Response 8. 100 g of cocoa beans were roasted to each time and temperature

Point 9. Table 2: The kinetic of catechin is not reported in the table; add in the text why.

Response 9. The kinetic parameters of the catechin are not shown because it corresponds to a combined production and degradation model, which does not correspond to Equations (2), (3) and (4). Therefore, we recommend further studies on this behavior of the catechin.

Point 10. Figure 1A and 1B are redundant with the data presented in Table 1 and, in some way, with kinetic parameters of Table 2 and can be deleted. Figure 1C reports only the fitting of 110°C, 190°C and 200°C: which was the reason of this choice?

Response 10: Figures 1A, 1B and 1C were deleted

Point 11. Figure 2: if R2 of the 90°C is excluded from the fitting, epicatechin is parallel to x axis and probably also TPC is not significant. The different pattern of 90°C is also evident in Table 1 and the Figure 2 can be deleted and text in the discussion reworded accordingly (Lines 242-244).

Response 12: Figure 2 was deleted. The text was reworded

Point 13. Conclusion: the authors did not measure the antioxidant activity and the sentence must be reworded.

Response 13: conclusion was reworded

Reviewer 2 Report

Despite not being particularly innovative, the topic of the manuscript could be potentially of interest in the field of cocoa derived products, especially in the framework of how roasting can affect the content of bioactive compounds that in turn affects both health and sensory properties of the final products.

Nonetheless, I do not think that in its present form the manuscript can be accepted for publication on Antioxidants.

Main Comments

Abstract

“The results showed a lower 23 degradation of TPC (13.95 ± 9.73%) and epicatechin (13.50 ± 18.10%)”. Are the Authors sure about the values reported? How can they justify such high standard deviations?

Introduction

Lines 60-64: I don’t think this is true; in some papers, cited by the authors themselves, kinetic parameters for polyphenols degradation (total procyanidins, epicatechin/catechin, …) upon roasting at different thermal conditions are reported. Please, check them and rephrase the sentence.

Materials & Methods

Can the Authors provide more information on the roasting process? How many beans (on a weight basis) were processed for each sample?

Results

Line 148: “the epicatechin concentration was 30% of the polyphenols”. How did the Authors calculate this value? If the percentage was obtained considering the content of epicatechin and catechin (HPLC) on total phenolic content (UV-Vis colorimetric determination), I don’t think it is correct.

Table 2: are such results correct? They do not seem so. Please, check them.

Table 3: such table reports values which are strictly connected to table 1. I would suggest to merge them.

Discussion

Results are poorly discussed and there is a lot of redundancy; most of the times, comments from the results section are repeated.

English language should be checked by a native speaker throughout the manuscript.

Reviewer 3 Report

In the manuscript, the authors studied kinetics of total phenolic content and monomeric flavan-3-ols during roasting process of Criollo cocoa. Overall, the data is interesting and solid. However, there are several concerns to be addressed:

Line 36-40: Please re-write the sentence. These is a grammaritic error. Line 59: ‘reported for’ should be corrected into ‘reported on’ Line 76-77: should be “according to their size’ and what is ‘taking care of a uniform size’? Line 82: ‘three cocoa beans were grinding’. Is it enough to represent the whole sample? Line 122: “means and standard deviations calculated ” should be “means and standard deviations were calculated” and What is “According with to”? Page 4 Table 1
The standard deviations are too large for some of the TPC (for example 200C 40min, 28.83 ± 28.90). Need to verify the methodology used for TPC quantification. Page 6 169-172
In Figure 1C, how about the fitting curve of other temperature other than 110, 190, 200?
The author may need to show and discuss the data of all roasting temperatures. Page 8 Table 3
The standard deviations are too large in table 3 especially for Catechin, which make the result less reliable. May need more repetition to confirm these data. Need to demonstrate and at least discuss in more detail what kind of thermal reactions take place upon roasting. What reactions are responsible for the changes of TPC, catechin and epicatechin?  How those reactions affect the concentration of these compounds under different temperatures?

Author Response

Response to Reviewer 3 Comments

Point 1. Line 36-40: Please re-write the sentence. These is a grammaritic error.

Response 1. We will request the English language correction service from the publisher

Point 2. Line 59: ‘reported for’ should be corrected into ‘reported on’

Response 2. This was corrected

Point 3. Line 76-77: should be “according to their size’ and what is ‘taking care of a uniform size’?

Response 3. This was corrected

Point 4. Line 82: ‘three cocoa beans were grinding’. Is it enough to represent the whole sample?

Response 4. This was corrected. It’s enough to each sample

Point 5. Line 122: “means and standard deviations calculated ” should be “means and standard deviations were calculated” and What is “According with to”?

Response 5. This was corrected

Point 6. Page 4 Table 1. The standard deviations are too large for some of the TPC (for example 200C 40min, 28.83 ± 28.90). Need to verify the methodology used for TPC quantification.

Response 6. The values were corrected

Point 7. Page 6 169-172. In Figure 1C, how about the fitting curve of other temperature other than 110, 190, 200?. The author may need to show and discuss the data of all roasting temperatures.

Response 7. All temperatures were discussed from figure 1

Point 8.

Page 8 Table 3. The standard deviations are too large in table 3 especially for Catechin, which make the result less reliable. May need more repetition to confirm these data. Need to demonstrate and at least discuss in more detail what kind of thermal reactions take place upon roasting. What reactions are responsible for the changes of TPC, catechin and epicatechin?  How those reactions affect the concentration of these compounds under different temperatures?

Response 8.

Its correct, it was corrected Discussions were corrected and improved

Round 2

Reviewer 1 Report

After the revision, the manuscript is suitable for publication. 

Author Response

no comments for reply

Reviewer 2 Report

The Authors have addressed most of the criticisms; however, some of them still need to be explained and commented.

The Response 1 to Point 1 cites “High standard deviations values may be due to a

roasting process that has not yet been standardized”. Can they explain what they mean? Are the Authors showing results on an experimental plan where the roasting process has not been standardized?

Table 2: it is really uncommon to present coefficients of determinations on a percentage basis. Since it is not a percentage of variance, I would suggest to express R2 in absolute terms.

Lines 194-195: This is a sentence that has been added in the R1 version; however, it reports a statement which may be critical.   How did the Authors calculate this value? If the percentage was obtained considering the content of

epicatechin and catechin (HPLC) on total phenolic content (UV-Vis colorimetric

determination), I don’t think it is correct.

Lines 195-196: “This difference in epicatechin content is mainly due to the cocoa variety and the anthocyanin content that it may contain, …”. How can the anthocyanin content affect and justify the difference in epicatechin content?

Lines 199-204: the epicatechin contents reported in this study are more at least one magnitude higher if compared to results reported in the literature by authors using comparable methods. This is quite surprising.

Author Response

Point 1. The Response 1 to Point 1 cites “High standard deviations values may be due to a roasting process that has not yet been standardized”. Can they explain what they mean? Are the Authors showing results on an experimental plan where the roasting process has not been standardized?

Response 1:

The sentence: “High standard deviations values may be due to a roasting process that has not yet been standardized” was a error. However, there are other authors who also obtained high standard deviations in the process of roasting cocoa beans. For example:

In: Hu et al. 2016. Physicochemical properties and antioxidant capacity of raw, roasted and puffed cacao beans. Food Chemistry 194. 1089–1094. The roasting of Forastero cocoa was performed at 190 °C for 15 min using an industrial coffee roaster. The TPC was 36.21 ± 5.44 (mg GAE/g sample defatted)

In: Å»yżelewicz et al. 2018. The effect on bioactive components and characteristics of chocolate by functionalization with raw cocoa beans. Food Research International 113 (2018) 234–244. Roasting process of Criollo cocoa of 15 min at 120 °C was performed. The (−)-Epicatechin was 192.31±2.01 to 361.70±5.22 (mg/100g)

In: Å»yżelewicz et al. 2016. The influence of the roasting process conditions on the polyphenol content in cocoa beans, nibs and chocolates. Food Research International 89. 918–929. Roasting process of Forastero cocoa of 15 min at 150 °C was performed. The catechin and epicatechin content was 20.81 ± 3.03 and 104.16 ± 14.52 mg/100 g, respectively.

Point 2. Table 2: it is really uncommon to present coefficients of determinations on a percentage basis. Since it is not a percentage of variance, I would suggest to express R2 in absolute terms.

Response 2: The error was corrected

Point 3. Lines 194-195: This is a sentence that has been added in the R1 version; however, it reports a statement which may be critical.   How did the Authors calculate this value? If the percentage was obtained considering the content of epicatechin and catechin (HPLC) on total phenolic content (UV-Vis colorimetric determination), I don’t think it is correct.

Response 3: The observation is correct, in that sense, the sentence was deleted

Point 4. Lines 195-196: “This difference in epicatechin content is mainly due to the cocoa variety and the anthocyanin content that it may contain, …”. How can the anthocyanin content affect and justify the difference in epicatechin content?

Response 4: This observation is congruent with the previous one (point 3); therefore, the sentence was deleted and the discussion was rewritten

Point 5. Lines 199-204: the epicatechin contents reported in this study are more at least one magnitude higher if compared to results reported in the literature by authors using comparable methods. This is quite surprising.

Response 5:

In another work done with Criollo cocoa (see table) from Perú (Multi-Service Cooperative APROCAM), dried fermented beans have been roasted in a tray dryer (Fischer Agro, Perú) at 110 °C for 2 hours. The epicatechin and catechin content (calculated by UHPLC) of the liquor obtained was 4.89 ± 0.00 and 0.75 ± 0.01 (mg/g defatted sample), respectively. The content of epicatechin and catechin in unroasted dried fermented beans was not measured. These values are still higher than those found in the literature. (The data has not yet been published).

Table. Functional properties of dark chocolate with incorporation of Arazá and its principal ingredients

Sample

Epicatechin (mg/gds)

Catechin (mg/gds)

Ingredient

Freeze-dried Arazá pulp (FDAP)

0.17 ± 0.01f

0.00 ± 0.00d 

Cocoa liquor

4.89 ± 0.00a 

0.75 ± 0.01a

Chocolate 

Control

3.22 ± 0.01d

0.60 ± 0.02b

Chocolate with 1% of FDAP

3.38 ± 0.01b

0.59 ± 0.01b

Chocolate with 1.5% of FDAP

3.33 ± 0.01c

0.60 ± 0.01b

Chocolate with 2% of FDAP

2.33 ± 0.00e

0.49 ± 0.01c

All values are shown as mean ± standard deviations (n = 3).

The different letters in the same column indicate significant differences (p<0.05).